# Rapid and Simultaneous Measurement of Fat and Moisture Contents in Pork by Low-Field Nuclear Magnetic Resonance

**DOI:** 10.3390/foods12010147

**Published:** 2022-12-27

**Authors:** Shuyue Tang, Yuhui Zhang, Wusun Li, Xiaoyan Tang, Xinyuan Huang

**Affiliations:** 1Key Laboratory of Agro-Product Quality and Safety, Institute of Quality Standard & Testing Technology for Agro-Products, Chinese Academy of Agricultural Sciences, Beijing 100081, China; 2College of Food Science and Technology, Nanjing Agricultural University, Nanjing 210095, China

**Keywords:** low-field nuclear magnetic resonance (LF-NMR), fat and moisture contents, measurement

## Abstract

In order to improve the efficiency of Soxhlet extraction and oven drying, low-field nuclear magnetic resonance (LF-NMR) technology was used to detect fat and moisture contents in pork. The transverse relaxation time (T_2_) distribution curves were constructed by Carr–Purcell–Meiboom–Gill (CPMG) experiments. In addition, the optimal conditions of adding MnCl_2_ aqueous solution was explored to separate water and fat signal peaks. Finally, the reliability of this method for the determination of fat and moisture contents in pork was verified. The present study showed that adding 1.5 mL of 20% MnCl_2_ aqueous solution solution at 50 °C can isolate and obtain a stable peak of fat. The lard and 0.85% MnCl_2_ aqueous solution were used as the standards for fat and moisture measurements, respectively, and calibration curves with R^2^ = 0.9999 were obtained. In addition, the repeatability and reproducibility of this method were 1.71~3.10%. There was a significant correlation (*p* < 0.05) between the LF-NMR method and the conventional methods (Soxhlet extraction and oven drying), and the R^2^ was 0.9987 and 0.9207 for fat and moisture, respectively. All the results proved that LF-NMR could determine fat and moisture contents in pork rapidly and simultaneously.

## 1. Introduction

Pork is rich in many nutrients, of which moisture and fat are not only the nutritional components but also the important indicators for evaluating the eating quality [1,2,3] and the safety quality of pork. Fat is one of the important nutrients, but excessive fat intake can cause adverse health effects. In addition, the content of fat directly affects the flavor, tenderness, and juiciness of the meat. Many biochemical reactions are closely related to water content, and the content of water also directly affects the quality of meat. Some studies showed that higher moisture levels could lead to more salmonella in meat [4]. Now, Soxhlet extraction and oven drying are conventional methods for the determination of fat and moisture contents, respectively. These methods are time-consuming and laborious, and large amounts of hazard organic reagents, such as ether or petroleum ether, are used in the evaluation process [5]. In order to avoid these short-comings, rapid non-destructive methodologies such as near-infrared spectroscopy [6,7], Raman spectroscopy [8,9], and hyperspectral imaging [10,11] technologies are used to evaluate pork quality. These technologies can accurately measure the moisture and fat contents of pork online, but stoichiometry methods and model building need to be applied for predictions with abundant samples. Moreover, the complicated data processing and high equipment cost are not conducive to popularization and application. Therefore, it is essential to establish a new method for the simultaneous and accurate determination of fat and moisture contents with high efficiency and low pollution.

In recent years, low-field nuclear magnetic resonance (LF-NMR) has been widely used to analyze and quantify the moisture and fat of fruits [12], grain [13], aquatic products [14], and meat [15]. In LF-NMR, the electromagnetic signal is generated by the perturbation of the sample core by an oscillating magnetic field. The specific proton signal is related to the absolute amount of hydrogen atoms. Bertram et al. [16] took the lead in studying the water distribution of pork by multi-component lateral relaxation of LF-NMR and finally concluded that bound water, difficult flowing water, and free water corresponded to the shortest, medium, and longest relaxation times of LF-NMR, respectively. This hypothesis theory has now become the theoretical basis of LF-NMR technology in food detection, and relevant research has been carried out. The moisture and fat contents were measured by hydrogen proton responses in magnetic fields and the transverse relaxation time (T_2_) distribution curves, which show the status and amount of the hydrogen proton. Some studies indicated that moisture and fat could not be distinguished by T_2_ distribution curves. Sørland et al. [17] proposed dry sampling for removing water to distinguish two T_2_ distribution curves. Wang et al. [18] illustrated that a small signal peak still existed in the same relaxation time period as water after drying, which was attributed to the hydrogen protons in the oil molecule. Miklos et al. [19] found that the signal peak in sausage at a T_2_ range of 60–160 ms contained both moisture and fat. Therefore, the key to determining the contents of fat and moisture is solving the overlap of the fat and moisture NMR signal and establishing the accurate relationships between fat and moisture weights and peak areas.

Many researchers were employed to reduce water content for resolving overlap, such as microwave drying, and they used LF-NMR to determine the fat content of dried samples. Monaretto used CPMG-CWFP-T1 to assign fat and water signals in beef samples. Keeton et al. [20,21] used this method to detect various meats, and the results showed good repeatability and reproducibility. In addition, Sørland et al. [17] used LF-NMR to determine fat and moisture contents in meat using two methods. One method was to evaporate water so that the water NMR signal was removed from the total NMR signal. The second method was using pulsed magnetic field gradients to separate the fat NMR signal from the water NMR signal. However, this method has high requirements on equipment and greatly increases the cost of detection. Paramagnetic ions, such as the manganese ion (Mn^2+^), can accelerate the decay process and reduce the T_2_ [22]. It was showed that adding Mn^2+^ to the oil sludge could obviously change the T_2_ distribution curves and separate the water and oil signal peaks [23]. The standard of the American Oil Chemists Society described the determination of solid fat content by low-resolution nuclear magnetic resonance, which can only determine the solid fat content and not the moisture content at the same time.

In this study, MnCl_2_ aqueous solution was used for the separation of moisture and fat LF-NMR signal peaks. Then, the sample treatment conditions, instrument parameters, and standard samples were explored to establish a safe, rapid, accurate, and simultaneous determination method of moisture and fat contents of pork, which provides a theoretical reference for future research.

## 2. Materials and Methods

### 2.1. Materials

Pork samples from different parts, including the longissimus dorsi muscle, plum blossom, streaky pork, and lard were purchased from a local market in Beijing, China. The MnCl_2_•4H_2_O and petroleum ether were chromatographically pure and were obtained from Shanghai Macklin Biochemical Co., Ltd. (Shanghai, China)

### 2.2. Pork Sample Preparation

The fasciae of pork were removed, and then the pork was ground into mince and stored in a refrigerator at −18 °C. MnCl_2_ aqueous solution was prepared with MnCl_2_•4H_2_O according to the different percentage content. The MnCl_2_ aqueous solution was added to 2.0 g of ground pork and mixed well, then stood at 50 °C for 30 min.

### 2.3. Effect of Sample Treatment Parameters on T_2_ Distribution Curves

#### 2.3.1. MnCl_2_ Aqueous Solution Concentration

1.0 mL of MnCl_2_ aqueous solution with different concentrations of 5%, 10%, 15%, 20%, 30%, 40%, 50%, and 60% were added into 2.0 g of minced meat, respectively, and then mixed and heated at 50 °C in a metal bath for 30 min.

#### 2.3.2. MnCl_2_ Aqueous Solution Volume

The 20% MnCl_2_ aqueous solution with different volumes of 0.5 mL, 1.0 mL, 1.5 mL, 2.0 mL, and 3.0 mL were added into 2.0 g of minced meat, respectively, and then mixed and heated at 50 °C in a metal bath for 30 min.

#### 2.3.3. Heat Treatment Temperature

An amount of 1.0 mL of 20% MnCl_2_ aqueous solution was added into minced meat and mixed well. Then, the samples were heated at different temperatures of 32 °C, 40 °C, 50 °C, 60 °C, and 70 °C in a metal bath for 30 min.

### 2.4. Fat and Moisture Contents Measurement

The Soxhlet extraction method (AOAC Official Method 991.36) and oven drying method (AOAC Official Method 950.46) were used to measure fat and moisture contents in pork samples, respectively, as the reference values.

### 2.5. LF-NMR Relaxation Measurements

The pork samples and standard samples were measured using an LF-NMR analyzer (NMI20-40V-I, Suzhou Niumag Analytical Instrument Co., Ltd., Suzhou, China), with a proton resonance of 20 MHz at 32 °C and a sample tube of 25 mm in diameter. The spin–spin transverse relaxation was measured using a sequence based on the Carr–Purcell–Meiboom–Gill (CPMG) sequence with the following parameters: 90° pulse width (P1) = 5.52 μs, 180° pulse width (P2) = 12.48 μs, waiting time (TW) = 2000 ms, and number of scans (NS) = 32. The T_2_ distribution curves were acquired by multi-exponential fitting with the simultaneous iterative reconstruction technique (SIRT) algorithm. The relaxation times, peak areas, and proportions of peak areas were recorded.

#### 2.5.1. Effect of Echo Time (TE) on LF-NMR Relaxation

An amount of 4.5 mL of 20% MnCl_2_ aqueous solution was added into 2.0 g of minced meat and mixed, and then the sample was heated at 50 °C in a metal bath for 30 min. LF-NMR measurements were performed at different TEs of 0.1, 0.15, 0.20, 0.25, 0.30, and 0.35 ms.

#### 2.5.2. Effect of Number of Echoes (NS) on LF-NMR Relaxation

An amount of 1.5 mL of 20% MnCl_2_ aqueous solution was added into 2.0 g of minced meat and mixed, and then the sample was heated at 50 °C in a metal bath for 30 min. LF-NMR measurements were taken at different NSs of 16, 32, 64, and 128 times.

### 2.6. Statistical Analysis

All experiments were carried out in triplicate. The statistical analyses were performed with SPSS 25.0 (SPSS Inc., Chicago, IL, USA). The differences among samples were analyzed by analysis of variance (ANOVA) and Duncan’s multiple range tests. Differences at *p* < 0.05 were considered significant. The results were presented as the mean ± standard deviation. Figures were created using Origin 2021 (OriginLab, Corporation, Northampton, MA, USA).

## 3. Results and Discussion

### 3.1. Effect of MnCl_2_ Aqueous Solution on T_2_ Distribution Curves of Lard and Pork Meat

The peak of the T_2_ curve within the relaxation time of 36–843 ms was ascribed to fat, whose proportions of the peak areas were between 87.37–98.92%. The iterations with different numbers were selected in multi-exponential fitting [24], but these peaks with different iteration numbers did not show obvious bimodal distribution as in other studies [25]. After adding 40% MnCl_2_ aqueous solution, the T_2_ curves were almost the same as those before adding the solution, and the peak areas of fat did not change significantly (*p* > 0.05). Since manganese salts are insoluble in hydrocarbon compounds, the addition of MnCl_2_ aqueous solution had no effect on the relaxation characteristics of hydrogen protons in the oil phase [26,27]. The T_2_ curve of pork was expressed by the black blocks in Figure 1. Three discontinuous peaks from left to right were ascribed to bound water (T_21_), immobile water (T_22_), and free water (T_23_), respectively, which was in accordance with other studies on the distribution of T_2_ in meat [15,28]. However, the peaks of fat and moisture were superimposed, and it was also impossible to be distinguished by T_2_ curves [29]. 

The T_2_ curve of pork with Mn^2+^ was represented by the gray dotted line in Figure 1. Two peaks, T_21_ and T_22_, were attributed to the hydrogen protons of water and fat, respectively. It can be seen that the T_23_ disappeared, T_22_ decreased greatly, and T_21_ increased, compared with the T_2_ curve of pork. Mn^2+^ can permeate into the sample through ion transport, resulting in direct contact between Mn^2+^ and the hydrogen protons of water. In addition, there was a strong spin–spin interaction between the spin magnetic moments of hydrogen protons and those of electrons in ions, which accelerated the decay of hydrogen protons in water [30,31]. Hence, the peaks of immobile water and free water decreased or even disappeared. Meanwhile, the hydrogen protons introduced by the MnCl_2_ aqueous solution enhanced the T_21_ peak area. Compared with the T_2_ curve of lard, the distribution of T_22_ was basically the same. It proved that adding the MnCl_2_ aqueous solution was feasible to solve the overlapping problem of moisture and fat signal peaks in pork. This method has been used to distinguish water–oil signals in crude oil [23].

### 3.2. Effect of Sample Treatment Conditions on T_2_ Distribution Curves

#### 3.2.1. MnCl_2_ Aqueous Solution Concentration

Figure 2a presents the T_2_ curves of the samples treated with different concentrations of the MnCl_2_ aqueous solution. With the increase in concentration, the T_21_ moved to the left, and the peak area was reduced gradually. When the concentration reached 10%, the distribution of T_21_ did not change significantly, but the peak area still gradually decreased. It was reported that the amount of paramagnetic substance had a great influence on spin relaxation [16]. The higher the concentration of Mn^2+^, the faster the decay of hydrogen protons in water, and the greater the relaxation signals loss [26]. Moreover, the T_22_ also manifested a significant left shift with the increase in MnCl_2_ aqueous solution concentration. When the concentration was lower than 10%, the starting relaxation time of T_22_ was 43.354 ms, which was significantly different from that of lard. When concentration was 20%, the shape of the peak and distribution of T_22_ were similar to that of lard. As the concentration continued to increase, the initial relaxation time of T_22_ phased down, and the shape of the peak gradually became wider and shorter. However, the peak area did not change significantly. While the concentration was 20%, the stability of the T_22_ peak area was good, and the relative standard deviation (RSD) was only 2.38%. It was suggested that 20% MnCl_2_ aqueous solution could separate water and fat signals better.

#### 3.2.2. MnCl_2_ Aqueous Solution Volume

The solution volume directly affects the quantity of Mn^2+^. Figure 2b illustrates T_2_ curves corresponding to the adding of different volumes of MnCl_2_ aqueous solutions. The peak area of T_21_ decreased with the augment of volume, and the shape of the T_22_ peak gradually became shorter and wider. The results were consistent with the influence of MnCl_2_ aqueous solution concentration. The distribution and peak area of T_22_ did not change significantly when the volume was more than 1.5 mL, and the peak area was significantly higher than the areas of the samples treated with 0.5 or 1.0 mL MnCl_2_ aqueous solution. Furthermore, the RSD of the T_22_ peak area decreased from 5.05% to 2.18%, indicating that the stability of the peak area of high-volume (≥1.5 mL) treated groups was better than low-volume (<1.5 mL) treated groups. Although the proportion of T_21_ reached 97.06% when adding a small volume addition (0.5 mL), there was a large deviation of T_22_ peak in multi-exponential fitting, which could lead to the uneven distribution of Mn^2+^ in the sample. Therefore, the solution volume was determined to be 1.5 mL, and then a stable peak area of fat was able to be obtained.

#### 3.2.3. Heat Treatment Temperature

Temperature is an important factor affecting the relaxation characteristics of samples. As shown in Figure 2c, the peak area of the T_21_ increased with the increase in temperature, and the T_22_ shifted to the right with the rise in temperature. At 40 °C, the T_22_ peak area was significantly higher than that of other temperatures (*p* < 0.05). Temperature can affect internal molecular structure and phase state. Lard was solid at room temperature, and it gradually melted into liquid after heating. In this process, the arrangement of the internal molecules of fat changed from order to disorder, and the binding force between molecules was weakened, resulting in a prolonged relaxation time. As the temperature continued to rise, the thermal movement of the system intensified, and the molecular fluidity increased [32], which led to the extension of T_2_. Lard can be completely melted at approximately 48.4 °C. Sørland et al. [17] heated minced meat to 70 °C to ensure that the fat phase was liquid. While vegetable oil existed in a liquid form at room temperature, generally, the sample temperature of vegetable oil was kept consistent with that of the magnet (32 °C) [33,34] in order to avoid the presence of both solid and liquid fat in the sample. A temperature of 50 °C was chosen as the sample temperature, at which the stability (RSD = 3.52%) of the fat peak area was good.

### 3.3. Effect of Performance Parameters on LF-NMR Relaxation

#### 3.3.1. Effect of TE on LF-NMR Relaxation

TE, as the interval time between echoes and should be quite short to minimize the diffusion effects. Figure 3 shows the T_2_ distribution curves of the samples detected at different TEs, where T_21_ represents water and T_22_ represents fat. The increase in TE caused T_21_ to move to the right, and the peak area obviously decreased. However, the T_2_ and peak area of T_22_ did not change significantly. This can be explained by the fact that increasing TE will cause molecular diffusion, which will lead to prolonged T_2_ [35]. Moreover, the diffusion of water is stronger than that of fat, and the hydrogen proton decay rate of moisture in Mn^2+^-treated samples is faster. Therefore, the relaxation behavior of T_21_ changed significantly, but the relaxation behavior of T_22_ did not.

The variation of TE within 0.10~0.35 ms affected the attenuation degree and relaxation interval of CPMG echo decay curves. When TE increased from 0.10 ms to 0.30 ms, the signal amplitude decreased from 257.42 to 57.31. As the TE continued to increase, the signal amplitude did not change significantly, which indicated that the signal had been completely attenuated. In addition, when TE was 0.30 ms, the RSD of the fat peak area was 3.06%. Therefore, in this study, TE was chosen as 0.30 ms.

#### 3.3.2. Effect of NS on LF-NMR Relaxation

Figure 4a shows the influence of different NSs on T_2_ distribution curves. With the increase in NS, the areas of the two relaxation peaks of T_21_ and T_22_ gradually increased. The peak area of T_22_ and NS had a good linear relationship. The equation was y = 64.41x − 5.0905, and R2 was 0.9999. The NS was proportional to the signal noise ratio (SNR), as shown in Figure 4b. When the NS was set from 16 to 128, the SNR increased from 184.82 to 290.44. There was no significant difference in SNR when NS was 32 or 64 times. However, if NS is further increased, the time of signal acquisition will increase. Meanwhile, the continuous excitation of a RF pulse will increase the sample temperature, which will affect the relaxation characteristics of the samples [36]. Therefore, the NS was set to 32 times, and the areas of T_22_ showed good stability (RSD = 1.28%).

### 3.4. Calibration Curves of Fat and Moisture in Pork

#### 3.4.1. Calibration Curves of Fat

In this study, lard was used as the standard sample to measure the fat content of pork. The T_2_ distribution curves of different weights (0.0114–0.6413 g) of lard were measured. As shown in Figure 5, the T_22_ peak areas were enlarged linearly with the increased weight of lard over a range of 0.0114–0.6413 g (equivalent to 0.55–39.99% of the fat content of 2 g of minced pork). There was a significant positive correlation between lard weight and T_22_ peak areas (*p* < 0.05), and the regression equation was: y = 34,313x + 71.172 (R^2^ = 0.9999). The signal generation in LF-NMR was related to the amount of hydrogen protons, which positively correlated with the lipid or moisture content in the samples [13,37]. We calculated the hydrogen proton density of lard by measuring fat peak areas, and then we measured the fat contents.

#### 3.4.2. Calibration Curves of Moisture

It was reported that using deionized water as a standard sample would result in greatly reduced detection efficiency, due to the excessively long relaxation time of water [23]. As a result, MnCl_2_ aqueous solution [38], a rehydratable freeze-dried meat cube [39], and CuSO_4_ aqueous solution [40] were used as standard samples for water quantification to improve detection efficiency. In this study, the MnCl_2_ aqueous solution was used as the standard sample to measure moisture. The fat peak area had been available from the T_2_ distribution curve of samples treated with 20% MnCl_2_ aqueous solution. Then, the moisture peak area was calculated using the following equation:$$A_{W}=A\frac{M_{f}}{M}-A_{f}$$
where *Aw* is the moisture peak area, *A* is the total peak area of minced fresh pork, *Mf* is the mass of the minced pork that was treated with 20% MnCl_2_ aqueous solution, *M* is the mass of minced fresh pork, and *Af* is the fat peak area.

The calculated hydrogen proton density of moisture in pork was 33,501. Figure 6a shows the correlation curve between the concentration of the MnCl_2_ aqueous solution and the peak area of moisture. With the increase in concentration, the peak areas decreased exponentially. According to the correlation curve in Figure 6a, 0.85% MnCl_2_ aqueous solution was required to simulate the hydrogen proton density of moisture in pork. Therefore, this solution was applied as the standard sample to construct the calibration curve of moisture.

The T_2_ distribution curves of different weights (0.4958–2.9867 g) of 0.85% MnCl_2_ aqueous solution were measured and are shown in Figure 6b. Only one peak (T_21_) appeared before 10 ms, and its peak point relaxation time was 0.73 ms. As the mass of solution increased, the peak area multiplied at the same rate. The solution weight was highly linear with the moisture peak area, and the linear equation was: y = 34,467.93x + 98.647 (R^2^ = 1.00). The T_2_ distribution of the standard sample (0.85% MnCl_2_ aqueous solution) was different from that of moisture in pork due to the complex states that existed of moisture in pork. However, it was suggested that the change of relaxation time did not affect the hydrogen proton density of water by comparing five different kinds of meat of livestock and poultry [41]. Therefore, it was feasible to use 0.85% MnCl_2_ aqueous solution as the standard sample.

### 3.5. Method Validation

The repeatability and reproducibility of three different samples (*n* = 6) were verified using the above method. The T_2_ distribution curves of pork with different fat contents were shown in Figure 7. The fat peak (T_22_) was basically coincident, and peak areas were stable. The RSDs of peak area were 2.04%, 1.57%, and 1.71%, respectively (Table 1).

Then, the intra-day and inter-day reproducibility of the method were validated in the morning, at noon, and in the evening on the same day and different days (day 1, day 3, and day 5). As shown in Table 2, the RSDs were 3.03% and 3.10%, respectively. The above results suggested that the method had good precision.

### 3.6. Determination of Fat and Moisture Content in Pork

The fat and moisture contents of 17 pork samples were determined using the Soxhlet extraction method and the oven drying method, respectively, with reference values contained at 1.94–36.61% and 47.33–73.02%. The peak areas of these samples were tested by LF-NMR, and then fat and moisture contents were calculated. Figure 8 shows the comparison of fat and moisture contents between the conventional method and the LF-NMR method, and the linear regression equation and R^2^ are exhibited. There was no significant difference in results between the two methods (*p* > 0.05). It can be observed that the accuracy of fat content (R^2^ = 0.9987) was better than moisture content (R^2^ = 0.9207) when using the LF-NMR method. The fat content measured by conventional method was higher than that measured by the LF-NMR method because the conventional method required prolonged heating, which led to the increase in fatty acid polarity and significantly increased fat weight, manifested as the increase in the peak area of fat [42]. Samples that were pretreated with MnCl_2_ aqueous solution were not affected by high temperature, and the measured values of fat by LF-NMR were more accurate.

However, the moisture content measured by the LF-NMR method was lower than the oven drying method. It could be attributed to the high sample temperature (50 °C) in our study, which resulted in the loss of partial moisture relaxation signals of fresh pork [43]. Han et al. reported that the total peak area of the samples decreased, and the peaks of immobile water moved left with the rise in temperature [44]. The binding force of moisture in different samples was varied, which led to a certain deviation in the determination of moisture content [45,46].

Therefore, compared with the conventional methods, the results of the LF-NMR method were more accurate, and the detection time was shortened to less than 3 min for a single sample.

## 4. Conclusions

In this study, LF-NMR was used to determine fat and moisture contents. In particular, the problem of overlapping signal peaks of moisture and fat in pork was solved by adding 20% MnCl_2_ aqueous solution with a volume of 1.5 mL at 50 °C. Lard and 0.85% MnCl_2_ aqueous solution were used as standard samples for determining fat and moisture contents, respectively. Furthermore, the results of the LF-NMR method showed a good correlation with conventional methods, with R^2^ being 0.9987 (fat) and 0.9207 (moisture), respectively. The present study indicated that LF-NMR could detect the fat and moisture contents of pork simultaneously and accurately.

## Figures and Tables

**Figure 1 foods-12-00147-f001:**
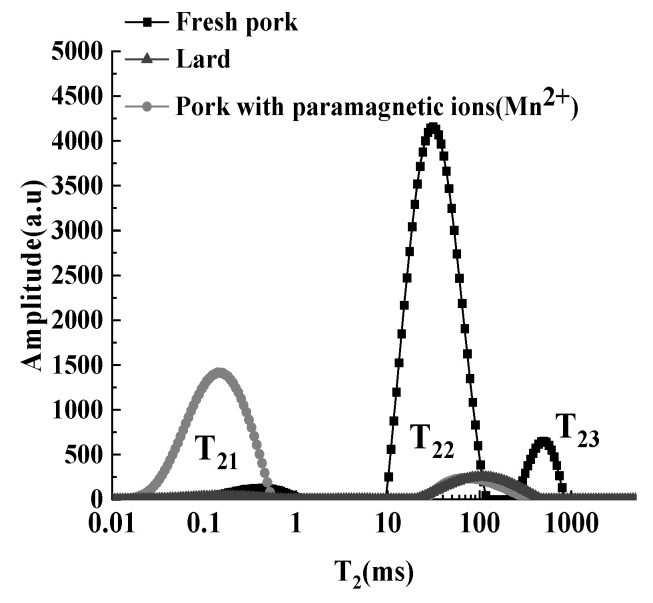
Effects of Mn^2+^ on T_2_ distribution curves of pork.

**Figure 2 foods-12-00147-f002:**
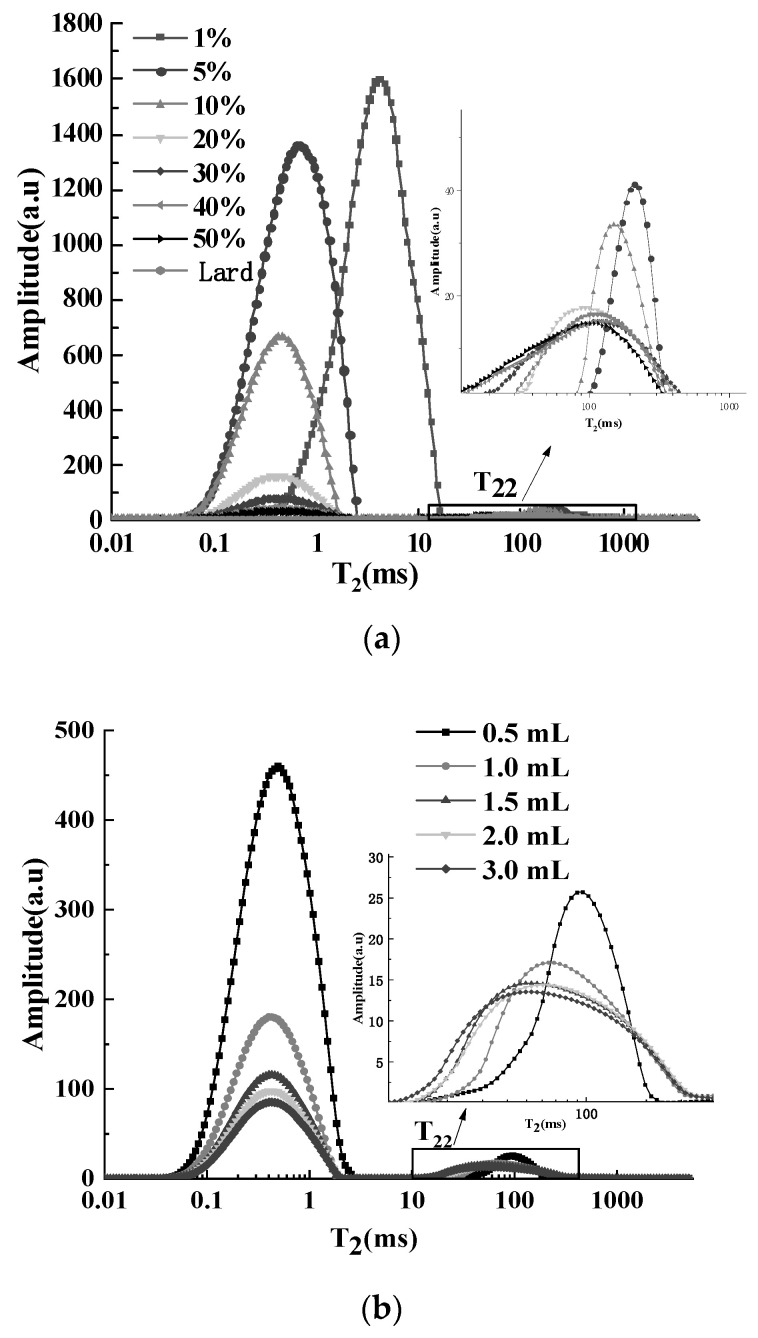
(**a**) Effects of MnCl_2_ aqueous solution concentration on T_2_ distribution curves of pork with Mn^2+^. (**b**) Effects of MnCl_2_ aqueous solution volume on T_2_ distribution curves of pork with Mn^2+^. (**c**) Effects of heat treatment temperature on T_2_ distribution curves of pork.

**Figure 3 foods-12-00147-f003:**
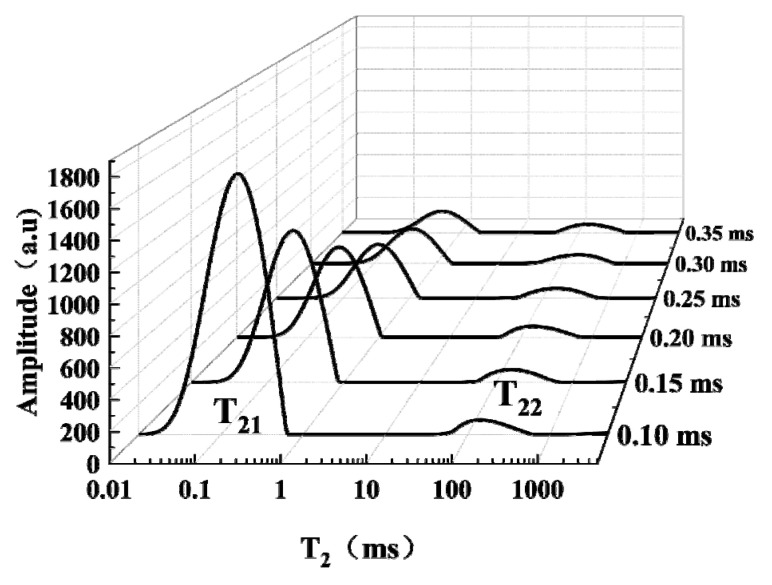
Effects of TE on T_2_ distribution curves.

**Figure 4 foods-12-00147-f004:**
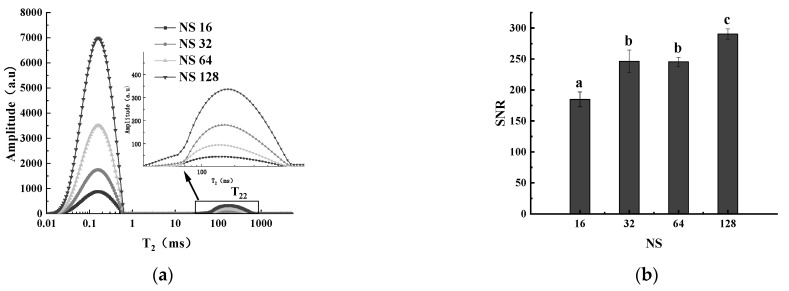
(**a**) Effects of NS on T_2_ distribution curves. (**b**) Effects of NS on SNR. Columns marked with different letters are significantly different according to Duncan’s multiple range test at *p* < 0.05.

**Figure 5 foods-12-00147-f005:**
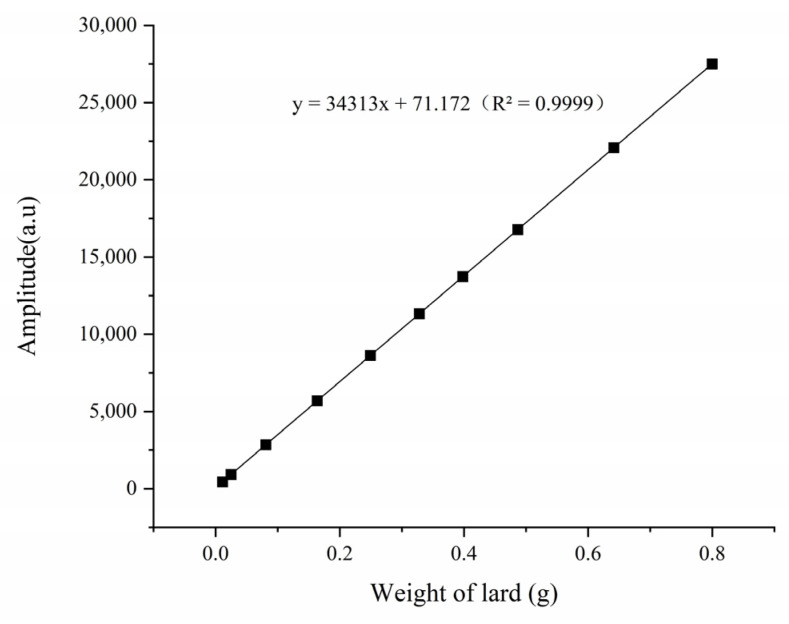
The calibration curves of lard with different weights and LF-NMR peak areas.

**Figure 6 foods-12-00147-f006:**
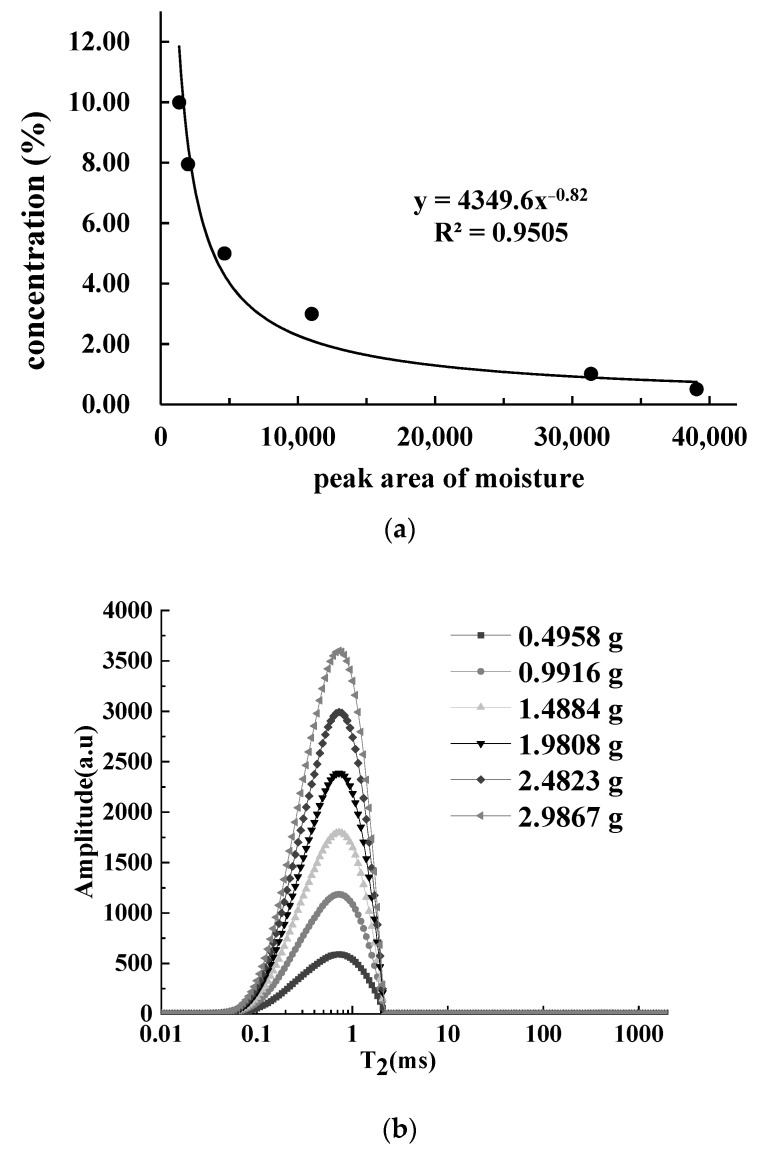
(**a**) Correlation between MnCl_2_ aqueous solution concentration and moisture peak areas. (**b**) T_2_ distribution curves of 0.85% MnCl_2_ aqueous solution with different weights.

**Figure 7 foods-12-00147-f007:**
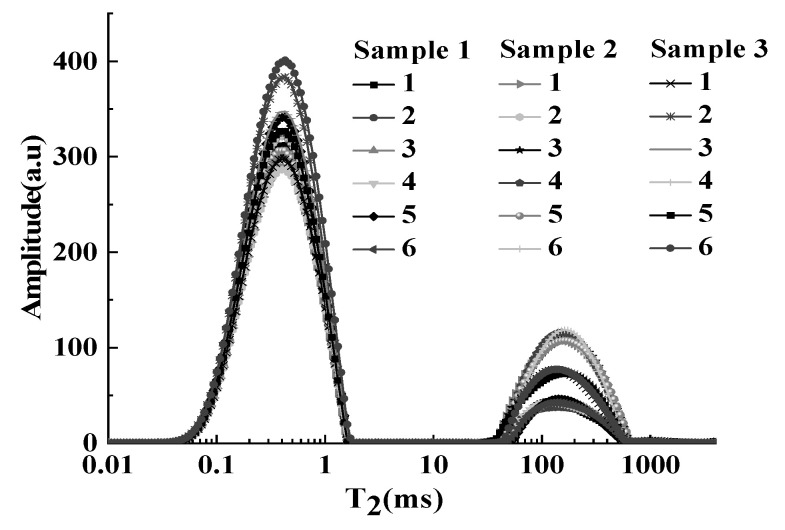
T_2_ distribution curves of pork with different fat contents.

**Figure 8 foods-12-00147-f008:**
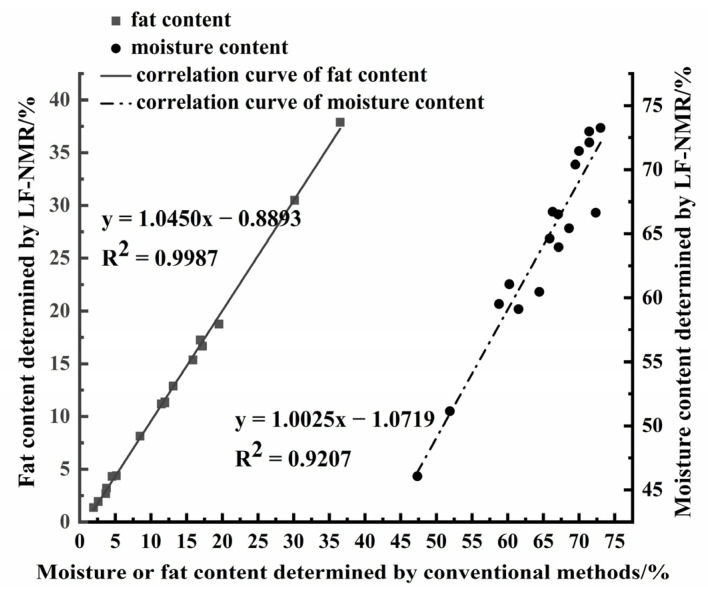
Correlation between reference values by conventional method and measured values by LF-NMR for fat and moisture contents of pork.

**Table 1 foods-12-00147-t001:** The validation of repeatability of fat peak areas.

Pork Sample	1	2	3	4	5	6	Fat Peak Area	RSD
1	1085 ± 18	1102 ± 14	1106 ± 15	1119 ± 6	1064 ± 11	1081 ± 18	1093 ± 22	2.04%
2	3068 ± 26	2971 ± 10	3036 ± 7	3062 ± 29	2991 ± 10	2956 ± 6	3014 ± 47	1.57%
3	2012 ± 23	2096 ± 10	2020 ± 11	2026 ± 19	2059 ± 13	2006 ± 10	2036 ± 35	1.71%

**Table 2 foods-12-00147-t002:** The validation of intra-day and inter-day reproducibility of fat peak areas.

	Intra-Day Reproducibility			Inter-Day Reproducibility		
	Morning	Noon	Evening	Day 1	Day 3	Day 5
	1081 ± 22	1050 ± 18	1067 ± 44	1492 ± 26	1398± 26	1455 ± 15
Fat peak area	1066 ± 32			1448 ± 45		
RSD	3.03%			3.10%		

## Data Availability

The data presented in this study are available upon request from the corresponding author.

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
