# Peer review of "Rapid and Simultaneous Measurement of Fat and Moisture Contents in Pork by Low-Field Nuclear Magnetic Resonance"

_foods, 2022, doi:10.3390/foods12010147_

Round 1

Reviewer 1 Report

I kindly accepted the invitation of review on the paper “Rapid and simultaneous measurement of fat and moisture contents in pork by low-field nuclear magnetic resonance”.

When animals are slaughtered, meat quality parameters are used as a tool to classify carcasses. Meat quality can be assessed in several ways, and the methods for doing this vary between people and over time. The main quality parameters of technological quality are water holding capacity, colour and fat. All these different meat quality parameters vary between animals, due to feed, sex, breed, housing etc.

The study aimed to investigate a efficiency of Soxhlet extraction and oven drying, the low field nuclear magnetic resonance (LF-NMR) technology  to detect fat and moisture contents in pork. The transverse relaxation time (T2) distribution curves were constructed by Carr-Purcell-Meiboom-Gill (CPMG) experiments. In addition, the optimal conditions of MnCl2•4H2O adding was explored to separate water and fat signal peaks. Finally, the reliability of this method for determination of fat and moisture content in pork was verified. The present study showed that adding 1.5 mL of 20% MnCl2•4H2O solution at 50℃ can isolate and obtain stable peak of fat. The lard and 0.85% MnCl2•4H2O solution were used as the standard for fat and moisture measurement, respectively, and calibration curves with R2=1.00 were obtained. In addition, the repeatability and reproducibility of this method were 1.71%~3.10%. There was a significant cor-relation (P < 0.05) between the LF-NMR method and the conventional methods (Soxhlet ex-traction and oven drying), and the R2 was 0.9987 and 0.9207 for fat and moisture, respectively. All the results proved that LF-NMR could determine fat and moisture contents in pork rapidly and simultaneously. 

In my opinion, the research is very straight-forward and clear. Introduction is informative and supported with relevant references. Methods are good described. The experiment, sampling, laboratory analyses and statistics were performed properly. Summary of results and discussion are interesting and are done in logical and reasonable manner. There are adequate number of  references.

Kind regards,

Author Response

Dear Reviewers:

Thank you for your comments on our manuscript, which are very helpful to improve our manuscript’s quality. 

Reviewer 2 Report

Pork has one of the most consumed meat in the world, however mostly quatitative parameters (like lean meat%) are used to describe its value in market. Eating quality of pork are not measured routinly, only for  some special market or breeding selection processes. Quick and relaiable methods for quantification of parameters (fat and moisture) related eating quality can facilitate the routine evaluation  of this trait in pig industry.

LF- NMR was used to determine fat and moisture content of pork samples. Several factors are tested to reach a highly reliable evaluation methods.

Title is clear just like the abstract of the manuscript. In the section introduction importance of this evaluation on pork should be exlain more (it is enclosed as comments in uploaded file). Material and methods are well detailed. Results are presented in several figures and tables, which was underlined and discussed with correspondent references.  However sometimes it is hard to understand or clearly distinguish the different treatments.

Conclusion is compact but comprehensive.

R2 should be changed to R2

Author Response

Dear Reviewers:

Thank you for your comments on our manuscript, which are very helpful to improve our manuscript’s quality. We have made modification according to your suggestions. Please see the cover letter uploaded for details. 

Reviewer 3 Report

The manuscript proposed a new LF NMR method to measure fat and moisture content in grounded pork. The method use a solution of Mn2+ to reduces water T2 and separate it from fat relaxation time.

In the abstract they reported R2=1. This is not observed experimentally unless you use only 2 or very few points.  It is practically impossible to have no interference from noise, instrumental distortions, operator errors, the reference method, etc. It would be better or more correct to use 0.9999 as the repeatability and reproducibility ranged from 1.7 to 3.1 %.  

 In the introduction, the authors reviewed the literature with only some of the papers that used NMR to determine the water and fat content in meat. I think that are other important papers must be included, mainly those by prof. Hanne Bertram who made the assignment of the relaxation time in pork. In addition, is necessary to add the AOCS method for determining the fat content in meat. Add also papers that use  T1 to measure the fat content in meat. According to these papers, the T1 values ​​of water and fat are naturally separated and there is a very simple equation for this purpose and even two-dimensional T1-T2 experiments

Some papers in this area. 10.1016/0309-1740(85)90078-6; 10.1111/j.1365-2621.1985.tb01899.x;10.1016/S0309-1740(01)00134-6.; 10.1016/s0309-1740(00)00080-2; 10.1016/j.foodcont.2013.10.026; 10.1016/j.jmr.2019.106666; 

In the manuscript the authors use MnCl2.H20. Is the manganese salt added directly in meat samples? It seems that they use an aqueous solution of MnCl2. Replace MnCl2.H20 with aqueous solution of MnCl2. 

Result and discussion.

 The T2 values ​​35,573 – 842,836. The error of T2 measurement is much larger than this resolution. Use only integer numbers. Also use integer number in all T2 and areas values and fat content determined by soxhlet ​​and use a decimal number for the percentages.

Figure 1a may be deleted.  Just inform in the text that T2 values ​​did not change int  the ranges that the Mn +2 ion was used.

Figure 2 caption: add. ....”curves of grounded pork, lard and pork with Mn2+.

Figure3. Only figure 3a is necessary. Figure 3b is the same result.

Figure 5 may be deleted. It is well known the dependence of NMR signal and sample mass

Author Response

(The authors gave the same response as above.)

Round 2

Reviewer 3 Report

The manuscript can be accepted in current version